# Improved kala-azar case management through implementation of health facility-based sentinel sites surveillance system in Bihar, India

**Vidya Nand Rabi Das**[1☯‡], **Niyamat Ali Siddiqui**[1☯‡], **Gouri Sankar Bhunia**[2], **Krishna Pandey**[1], **Sanjay Kumar Sinha**[1], **Md Zahid Ansari**[1], **Roshan Kamal Topno**[1], **Chandra Sekhar Lal**[1], **Alok Ranjan**[3], **Vijay Pratap Singh**[1], **Pradeep Das**[1,4]*

1 ICMR-Rajendra Memorial Research Institute of Medical Sciences, Patna, India, 2 Randstad India Private Limited, New Delhi, Delhi, India, 3 All India Institute of Medical Sciences, Patna, India, 4 Indira Gandhi Institute of Medical Sciences, Patna, India

☯ These authors contributed equally to this work.
‡ These authors share first authorship on this work.
* drpradeep.das@gmail.com

**Data Availability Statement:** Data Availability Statement The data that support the findings of this study are available from the Head, library and

## Abstract

### Background

Visceral leishmaniasis (VL), also known as kala-azar (KA), is a neglected vector-borne disease, targeted for elimination, but several affected blocks of Bihar are posing challenges with the high incidence of cases, and moreover, the disease is spreading in newer areas. High-quality kala-azar surveillance in India, always pose great concern. The complete and accurate patient level data is critical for the current kala-azar management information system (KMIS). On the other side, no accurate data on the burden of post kala-azar dermal leishmaniasis (PKDL) and co-infections are available under the current surveillance system, which might emerge as a serious concern. Additionally, in low case scenario, sentinel surveillance may be useful in addressing post-elimination activities and sustaining kala-azar (KA) elimination. Health facility-based sentinel site surveillance system has been proposed, first time to do a proper accounting of KA, PKDL and co-infection morbidity, mortality, diagnosis, case management, hotspot identification and monitoring the impact of elimination interventions.

### Methodology/principal findings

Kala-azar sentinel site surveillance was established and activated in thirteen health facilities of Bihar, India, using stratified sampling technique during 2011 to 2014. Data were collected through specially designed performa from all patients attending the outpatient departments of sentinel sites. Among 20968 symptomatic cases attended sentinel sites, 2996 cases of KA and 53 cases of PKDL were registered from 889 endemic villages. Symptomatic cases meant a person with fever of more than 15 days, weight loss, fatigue, anemia, and substantial swelling of the liver and spleen (enlargement of spleen and liver).The proportion of new and old cases was 86.1% and 13.9% respectively. A statistically significant difference was

information officer of RMRIMS, upon reasonable request. Details of the Library and Information Officer Name: Sri Brij Nath Prasad Head of the Dept. (Library & Information) Email: prasad.brijn@icmr.gov.in Ph.: +91-612-2633367 Fax: +91-612-2634379 https://www.rmrims.org.in/Library&InformationCentre.html.

**Funding:** The study was funded by National Vector Borne Disease Control Programme (NVBDCP) under World Bank Scheme for vector borne disease, Ministry of Health and Family Welfare, Govt. of India (Ref No: 3-87/2009-NVBDCP (KA)/WB/RMRI/Sentinel surveillance). Diagnostics and Drugs used for this study were provided by NVBDCP under National Kala azar Control Program. However, adequate and appropriate staff and prescribed treatment cards were provided by the state programme office through respective district program at the sentinel sites. Institutional support was also received for various project related activities. The funders had no role in study design, data collection, analysis, Preparation of the manuscript and decision to publish.

**Competing interests:** The authors have declared that no competing interests exist.

observed for reduction in KA incidence from 4.13/10000 in 2011 to 1.75/10000 in 2014 ($p < 0.001$). There were significant increase (0.08, 0.10 per 10 000 population) in the incidences of PKDL and co-infection respectively in the year 2014 as compared to that of 2011 (0.03, 0.06 per 10 000 population). The proportion of HIV-VL co-infection was significantly higher (1.6%; $p < 0.05$) as compared to other co-infections. Proportions of male in all age groups were higher and found statistically significant (Chi-square test = 7.6; $P = 0.026$). Utilization of laboratory services was greatly improved. Friedman test showed statistically significant difference between response of different anti kala-azar drugs (F = 25.0, P = 0.004). The initial and final cure rate of AmBisome was found excellent (100%). The results of the signed rank sum test showed significant symmetry of unresponsiveness rate ($P = 0.03$). Similarly, relapse rate of sodium antimony gluconate (SAG) was also found significantly higher as compared to other drugs (95%CI 0.2165 to 19.7035; $P = 0.03$). A statistically significant difference was found ($p < 0.001$) between villages having 1–2 cases (74%) and villages with 3–5 cases (15%). Significantly higher proportion (95%) of cases were captured by existing Govt. surveillance system (KMIS) ($p < 0.001$), as compared to private providers (5%).

## Conclusions/significance

Establishment of a sentinel site based kala-azar surveillance system in Bihar, India effectively detected the rising trend of PKDL and co-infections and captured complete and accurate patient level data. Further, this system may provide a model for improving laboratory services, KA, PKDL and co-infection case management in other health facilities of Bihar without further referral. Program managers may use these results for evaluating program's effectiveness. It may provide an example for changing the practices of health care workers in Bihar and set a benchmark of high quality surveillance data in a resource limited setting. However, the generalizability of this sentinel surveillance finding to other context remains a major limitation of this study. The justifications for this; the sentinel sites were made in the traditionally high endemic PHC's. The other conditions were Program commitment for diagnostic (rk-39) and the first line anti kala-azar drug i.e. miltefosine throughout the study period in the sentinel sites. In addition, there were clause of fulfillment of readiness criteria at each sentinel site (already described in the line no 171 to 180 at page no-8, 181–189 at page no-9 and 192–212 at page no-10). Rigorous efforts were taken to improve all the sentinel sites to meet the readiness criteria and research activities started only after meeting readiness criteria at the site. Therefore sentinel site surveillance described under the present study cannot be integrated into other set up (medium and low endemic areas). However, it can be integrated into highly endemic areas with program commitment and fulfillment of readiness criteria.

## Author summary

Visceral leishmaniasis is a neglected vector-borne disease, and one of the major public health problems of Bihar. The disease has been targeted for elimination by 2020. Bihar still posing challenge i.e. incidence is much high in a number of affected blocks, and even the disease is spreading in newer areas. None availability of an accurate data on the

burden of post kala-azar dermal leishmaniasis (PKDL) and co-infections under the current surveillance system may emerge as a serious concern. Therefore, health facility-based sentinel site surveillance system has been attempted for the first time in Bihar for proper accounting of KA, PKDL and co-infection morbidity, mortality, diagnosis, case management, hotspot identification and monitoring the impact of elimination interventions. A system for capturing regional transmission is essential that can target focal areas of infection to monitor progress towards kala-azar elimination. Kala-azar sentinel site surveillance was established and activated in thirteen health facilities of Bihar during 2011 to 2014. The results showed a significant increase in PKDL and co-infection in 2014 when compared to 2011. Findings further revealed that utilization of laboratory services and case management were greatly improved, as majority of patients with KA, PKDL & co-infections were managed by the sentinel sites itself. The final cure rate of AmBisome was found excellent (100%). These observations may be useful to provide the basis for the design, refinement and resource allocation of the kala-azar control program. This system may also be useful in measuring impact of elimination interventions, their effectiveness and finally help in improving program management. It may further be used as an example for changing the practices of health care workers in Bihar and a lesson how to achieve high quality surveillance data in a resource limited setting. Standardization of sentinel site surveillance in terms of improved procedure, training, logistics, etc may further increase the effectiveness of this system. It may possibly be used as a cost-effective approach for capturing real-time kala-azar data under national kala-azar elimination programme.

## Introduction

Visceral leishmaniasis (VL) is a neglected vector-borne disease caused by *Leishmania donovani* parasite and is transmitted by the bite of infected sand flies *(Phlebotomus argentipes)*. The annual burden of VL cases worldwide is 0.2±0.4 million per year. The majority of the cases occur in five countries viz. India, Bangladesh, Sudan, Ethiopia and Brazil. Out of the worldwide VL cases, about 50% reported from India alone, however, Bihar state contributes 80% of VL cases. Most of the endemic population is poor from rural areas living in mud houses [1]. It is suspected to be underestimated by the routine health programme statistics in the absence of a robust surveillance mechanism [2–4]. In Bihar, leishmaniasis manifests in two forms i.e. KA and PKDL. Post kala-azar dermal leishmaniasis is a secondary complication of visceral leishmaniasis (VL) and non-fatal [5]. The disease has been targeted for elimination (reducing incidence to less than one per 10,000 populations at sub-district level by 2020) in the Indian subcontinent due to the availability of field based diagnostics (rK-39 strip test) and effective drugs [6]. Surveillance is a key component for any disease control programme but unfortunately in India, kala-azar surveillance is weak. Few patients still seek treatment in the private sector and are not tracked for treatment, efficacy and compliance in the kala-azar management information system (KMIS). Effective kala-azar surveillance may be helpful in program planning and management. It further informs governments and donors about progress towards kala-azar elimination. These may also be useful for the design, refinement and resource allocation of control programs [7, 8]. Kala-azar elimination program in India mainly depends on routinely collected health facility-based data through KMIS. The data collected at health facilities vary widely and are subject to bias. There are uncertainties, that whether a patient with kala-azar has been captured by this system correctly. The disease burden of PKDL and co-infections are not well known under current surveillance system. These are emerging as a serious concern

and are reported from 36 endemic countries including Bihar, India. There is a need to establish strong surveillance for both conditions including kala-azar. On the other end, active case detection would become important to kala-azar elimination. Currently, kala-azar fortnights are being organized for active case detection at the village level, but it is very expensive and resource intensive.

The current capacity of routine VL surveillance is inadequate and indicators are not properly defined. The quality of data reporting and completeness are also not adequately addressed. Timeliness reporting is also variable. The capacity of data analysis, interpretation, and action is extremely limited. The diagnosis of PKDL and co-infections is of variable quality or absent as in the case of malaria [9, 10]. However, disease surveillance is considered as one of the principal activity of public health systems. It is always expansive and time taking activity [11]. Kala-azar case reporting is present in endemic districts across the state. A provision of accurate and timely VL surveillance is essential for elimination as in the case of malaria [11]. A well defined system is much required to document regional transmission to monitor progress of kala-azar elimination programme. An effective surveillance system is warranted to target focal areas of infection, and increase capacity to identify hotspots of disease transmission. This may be helpful in monitoring real-time kala-azar data and in identifying changes in kala-azar transmission, morbidity and mortality.

Sentinel surveillance approach for any disease may be useful for early detection of outbreaks, measuring disease burden, change in morbidity, mortality patterns and for timely implementation of control and preventive measures. This is a system in which a designated group of reporting sources such as hospitals; agencies agree to report all cases of one or more notifiable conditions. Primarily, health facility-based sentinel site surveillance system may be useful to obtained disease trends and improving case management. Sentinel surveillance offers greater design flexibility as in the case of malaria and tuberculosis, when compared to population-based surveillance [12, 13]. Although results obtained through sentinel surveillance may not be representative of the entire population but it may be useful in reflecting the trends based on accurate assessment. Similar studies on kala-azar are not available in Bihar. The major objectives of the study were proper accounting of VL, PKDL and co-infection morbidity, mortality, diagnosis, case management, hotspot identification and monitoring the impact of elimination interventions. The other objectives were monitoring of real-time kala-azar and PKDL data for assessing changes in kala-azar/PKDL/Co-infection morbidity, mortality and pattern of disease transmission. In addition, responses to different anti kala-azar drugs administered at sentinel sites were also evaluated. Present study was aimed to describe the design and implementation of kala-azar surveillance and its utilization for epidemic detection and response. Further, it may be used for programme decision making. Data collected under this study may fill the gap of existing KMIS and reassess the prioritization of resources. The information emerging from the kala-azar management information system (KMIS) can also be validated. Further, it may be useful in measuring the impact of elimination interventions. To the best of our knowledge, this is the first study of its type from Bihar on kala-azar.

## Materials and methods

### Ethics statement

The study was duly approved by the Rajendra Memorial Research Institute of Medical Sciences, Ethics Committee and the Scientific Advisory Committee. A written informed consent was obtained from the head of all sentinel sites. A written informed consent was also obtained from the subjects admitted for treatment at sites. Subjects <18 years old who were able to read provided written assent following written consent from their parent or guardian. All study

data were anonymized before analysis. Confidentiality was maintained in full compliance with the principles of the Declaration of Helsinki (as amended in Tokyo, Hong Kong, and Somerset West, South Africa).

## Study area and selection of sentinel sites

In Bihar, twenty one (21) districts were identified as high endemic for kala-azar for the year 2010, reporting on an average 1800 cases, in a range of 208–4660 cases. The incidence proportion ranged from 1 to 25 per 10 000 persons with an average of 7 per 10 000 persons (kala-azar incidence data provided by State Programme Office, Bihar). All these districts are situated mainly to north of the river Ganges. Nearly 80% of the total cases reported by the programme were from 12 districts of Bihar. These districts were East Champaran, Sitamarhi, Madhubani, Darbhanga, Muzaffarpur, Samastipur, Vaishali, Saran, Begusarai, Saharsa, Madhepura and Purnia. Sentinel sites were made in 4 high endemic districts of Bihar, namely, Muzaffarpur, Samastipur, East Champaran and Saran as per their incidence proportion rate (IPR). These districts were selected because of piloting of various interventions under the existing kala-azar elimination program, and on the basis of the geographical locations. The districts were selected from all the direction. The due consideration has been given to Program commitment for making provision of rapid diagnostic test kits (rK-39) and the first line anti kala-azar drug i.e. miltefosine in the selected study districts. In each district, two primary health centers (PHC's) were selected as sentinel sites again as per the incidence proportion rate (IPR) (Fig 1). Fig 1 has been modified from the adopted map from the public domain (www.bihar.gov.in/http://www. maps of india.com) as per requirement. Additionally, one district hospital from each district and one medical college, located at Muzaffarpur were selected as sentinel sites. Thus the present study was conducted in thirteen sentinel sites. (Fig 1).

## Launching of health facility-based sentinel sites

ICMR-Rajendra Memorial Research Institute of Medical Sciences (ICMR-RMRIMS), Patna in collaboration with National Vector Borne Diseases Control Programme (NVBDCP), State Program office as well as District Program office, Bihar established a health facility based kala-azar sentinel surveillance sites. These sentinel sites were selected as per the fulfillment of readiness criteria viz.

i. Availability of adequate and appropriate staff at the selected sentinel sites.

ii. Trained and competent project staff with good understanding of their roles and responsibilities for the research work.

iii. Motivated health care providers with package of services for kala-azar.

iv. Consistent availability of diagnostics, drugs, equipment etc.

v. Provision of individual drug packet for each diagnosed kala-azar case to ensure complete treatment.

vi. Availability of treatment cards and its accurate filling by the health care providers at the study site.

vii. Regular motivational activities for kala-azar patients by the health workers to complete their treatment.

viii. Operational follow-up mechanisms in place at the site.

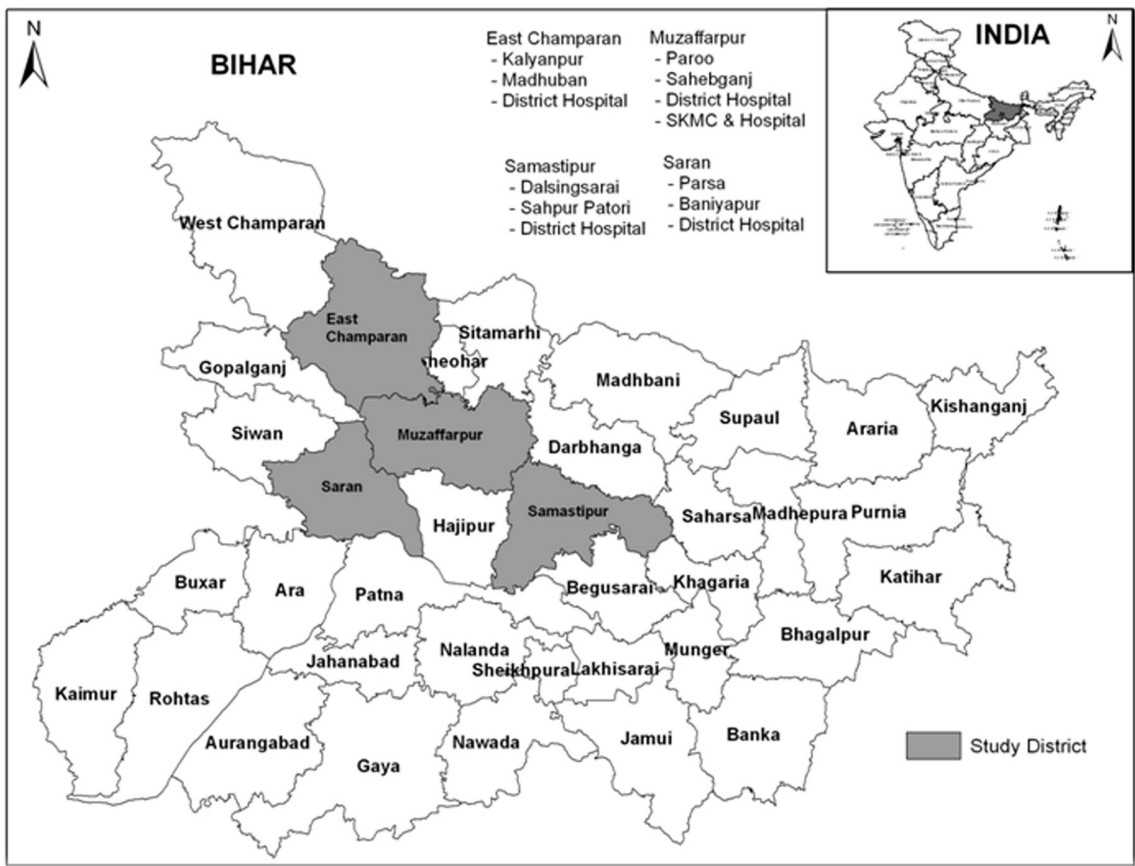

**Fig 1. Location map of the study area (www.bihar.gov.in/http://www.maps of india.com).**

Deficiencies of all selected sentinel sites were identified. Rigorous efforts were taken to improve all the sentinel sites to meet the readiness criteria within stipulated times (3–6 months). Research activities were started after meeting readiness criteria at the site.

## Sample size

Keeping the criteria of selection of sentinel site, fulfillment of readiness criteria for health facility based sentinel sites and resources in mind; the sample size for this study was restricted to 13 sentinel sites.

## Sampling technique

Study districts were selected on the basis of piloting of various interventions under the existing kala-azar elimination program, and on the basis of the geographical locations. In addition, due consideration has been given to program commitment for making provision of rapid diagnostic test kits (rK-39) and the first line anti kala-azar drug i.e. miltefosine in the selected study districts. The stratified sampling technique was attempted for the selection of sentinel sites according to the incidence proportion rate (IPR) of PHC's.

## Study duration

The present study was conducted during the period 2011 to 2014.

## Recruitment of personnel for sentinel sites

Four Site Supervisors (Research Officer) with post-graduate qualification and working experience in public health were appointed for all the study districts. The work of one district was assigned to one site supervisor. These site supervisors were imparted extensive two-weeks training at RMRI related to kala-azar management and surveillance techniques.

## Training of site supervisors and quality assurance

Staff including site supervisors at the sentinel sites were imparted two weeks extensive training covering all important aspects of kala-azar in order to enable them to discharge their responsibilities smoothly. Training was organized at RMRI where senior faculty and experts were engaged as the facilitators/trainers. The curriculum and detailed training module in form of written text material and standard operating procedure (SOP) was prepared. Training module was developed by the senior faculties of RMRI in consultation with the NVBDCP, keeping in view the desired job responsibilities and activities of the site supervisors and other staff engaged. During the training, all the procedures were taught through hands-on training using the formats proposed to be used. Besides, the induction training, RMRI also organized on-site training periodically and regularly throughout the project tenure to the site supervisors and other staff members involved in sentinel sites activities for improving skill and motivating the staff members. They were imparted short-term hands on (recurrent) training on various aspects like patients card filling, data management, computer skills, data presentation and reporting etc. A hand on practice was also conducted under supervision of the senior faculties of RMRI including principal investigator.

## Sentinel surveillance network

There were two types of network agencies and functionaries working at different level separately and in close coordination with other agencies. First networking worked on the basis of hierarchical system using bottom-to-top approach in terms of responsibilities and accountabilities. A "Memorandum of Understanding (MoU)" was signed by all the participating sentinel sites identified in the surveillance network. The sentinel surveillance is limited to level-II and level-III facilities only because of the availability of diagnosis and treatment at these level only. The involvement of level-I facility staff was limited to early detection of VL cases and for follow up of patients, and for regular monitoring of adverse effects. Every sentinel site was assigned the task of completion of treatment for each patient. None of the private health care facilities dealing with kala-azar was inducted under the sentinel sites for this study due to lack of evolved cost effective mechanism. The criteria to utilize the services of existing facilities were developed by the site supervisors with the help of kala-azar technical supervisors (KTS's) and vector borne diseases consultants (VBD's) of the district. RMRI was taken special effort to ensure that these centers must satisfy all the readiness criteria and comply with the norms and standard required for a sentinel site, before signing MoU. After the signing of memorandum of understanding the staff was imparted short training for filling up the patient cards and keeping the records complete in terms of treatment, side effects and follow-up to assess the effectiveness of drugs used for treatment.

## Case definition, diagnosis and case management

A standard protocol for the case definition of KA, PKDL, co-infection, and severe case of KA, diagnosis, and case management was adopted by all the sites under the surveillance network for maintaining uniformity in recording and reporting across the sentinel sites.

## Case definition of kala-azar

An individual with fever of more than 15 days duration with splenomegaly and not responding to anti-malarial drugs and found positive by rapid diagnostic blood test (rK-39 strip test), or a case confirmed with laboratory-based diagnosis i.e. splenic/ bone marrow aspirate examination for *Leishmania donovani* (L.D.) {Accelerated plan for Kala-azar Elimination, 2017, WHO/ NVBDCP}.

## Case definition of post kala-azar dermal leishmaniasis (PKDL)

An individual with or without past history of kala-azar presented with maculo-papular or nodular lesions with rK-39 test positive, confirmed by slit-skin smear / skin-biopsy examination for presence of parasites was labeled as PKDL case {Accelerated plan for Kala-azar Elimination, 2017, WHO/NVBDCP}.

## Case definition of co-infection

A confirm case of kala-azar with *human immunodeficiency virus* (HIV)/tuberculosis (TB) positive test result was declared as co-infection.

## Severe case of kala-azar

The definitions used for severe cases of kala-azar were:

- Resistant and relapse cases of kala-azar.
- Kala-azar with severe anemia (Hemoglobin less than 5 g/dL).
- Kala-azar patient suffering from another serious disease like renal and hepatic conditions.

## Diagnosis of KA and PKDL

A standard protocol for the diagnosis of KA and PKDL was followed by all sentinel sites. The case of kala-azar was either diagnosed by positive test result of rapid diagnostic strip test (rK-39) or with laboratory-based diagnosis i.e. splenic/ bone marrow aspirate examination for *Leishmania donovani* (L.D). PKDL case was diagnosed by rK-39 positive test result and by slit-skin smear / skin-biopsy examination for presence of parasites {Accelerated plan for Kala-azar Elimination, 2017, WHO/NVBDCP}. The rapid diagnostic test kit (rK-39) of kala-azar and PKDL was made available in all the sentinel sites by the site supervisor.

## Diagnosis of co-infection and severe cases

Co-infection cases were diagnosed with the help of sputum for AFB/RDT for tuberculosis, X-ray chest, ELISA (TB) and ELISA (HIV). Additional tests viz. hematogram, biochemical, etc. examinations were also done to capture co-infection. However, such cases were finally referred to RMRIMS for diagnosis and treatment.

## Case management of KA and PKDL

All the sentinel sites followed a standard protocol of the treatment for kala-azar and PKDL. Treatment of kala-azar and PKDL was provided after admitting the patient at sentinel sites. The complete information of hospitalized cases was recorded in the patient treatment card. Treatment cards were in place at every sentinel site. Vital information on the place, age, sex, marital status and pregnancy and its outcome was filled up in the treatment card. Additionally, information about drug used for treatment of KA & PKDL, treatment duration, treatment status, adverse events of drugs, reasons of treatment discontinuation of treatment other than side effects of drug, etc., were also collected.

Site Supervisors in each district motivated the treating physicians and other staff members to fill up the form completely. The data was collected at each site with the help of standard operating procedure (SOP), used for filling various formats to ensure the quality of data. Filled up patient treatment cards were collected by the site supervisor at each site. The filled up forms were checked and verified by the site supervisors who filled the pre-designed template form and create patient database for their respective districts. Line listing of kala-azar, PKDL and disaggregated reports were generated every month using database.

In case of death in course of treatment and follow-up, causes of death were recorded by the site supervisor with the direction of the treating physicians. The causes of death were investigated thoroughly to assess the specific cause of deaths such as drug intolerance, toxicity, cardiac-arrest and other reasons for deaths in detail. The classification of death after patient review and audit was identified as a death attributable to kala-azar or death that caused with an associated illness. A list of patients who died during the course of treatment or follow up was prepared monthly. The treating physicians at each sentinel sites was directed to certify and fill-in the cause of death column in the patient treatment card properly. Verbal autopsy was performed by a trained person. Village health workers and volunteers engaged in this project working at sub-PHC were instructed to report every death that occurred in their areas of monitoring to the sentinel site supervisors. RMRIMS constituted a panel of experts to review the report of the verbal autopsy conducted by the site supervisors and coordinator. In any circumstances, cause of death was finally decided by the panel of experts constituted for the purpose. Sentinel sites adopted above defined protocol for setting an example to others, not functioning under surveillance network. The informations on verbal autopsy of the cases not admitted to any hospital were tracked and reported separately. Kala-azar patients admitted to the sentinel sites were used for estimation of case fatality rates of the disease.

## Case management of co-infection

Treatment of co-infections was provided as per the guidelines of NVBDCP, AIDS control programme and TB control programme. The testing for HIV of kala-azar patients at high risk was performed at district level. However, such cases were finally referred to RMRIMS for final diagnosis and treatment.

## Case management of severe cases of KA

Severe anemic patients were treated with blood transfusion and when hemoglobin raised by medicines (iron and folic acid), specific treatment was started.

## Data management and information system

Records of all the patients visiting and enrolled for treatment at the sentinel sites were collected in a pre-designed format. All the formats used under the study were pre-tested and modified

accordingly. Site supervisors collected completely filled patient cards and other records for all the patients. Checking and verification of all filled in patient cards/records to maintain the consistency in data were performed by coordinator of the study. Incomplete patient's card and errors found in cards were re-checked and verified with the assistance of the physicians at the sentinel sites for immediate corrections. Completely filled in and checked cards were transferred to the computer using a pre-designed electronic template for creating database for each site and handed a copy of the monthly data to the district program managers such as VBD consultant and District VBD Officer. Checking and verification of the entire data sheet in the database was done by District VBD consultant at district level and feedback on report if any was sent to site supervisor present in the district. The site supervisors were also trained to analyze and disseminate the data in form of graphics and reports to higher levels. At the same time, information for action at each level was used for feedback and for capacity development, which helps in improving the services rendered to the patient. All the information from each of the sentinel sites were shared with RMRI. RMRI consolidates the data and generate the report which sent to all higher levels in the existing system and feedbacks from higher levels were received, taken into consideration for improvement in future report. Every effort was taken to strengthen the information collected and thus to improve existing surveillance system. Special efforts were taken to monitor the quality of data generated at these sentinel sites. A copy of the monthly report was sent by site supervisor to the district VBD officers for appropriate action and to provide feedback on monthly report. A consolidated report from sentinel sites pertaining to the major achievements for kala-azar was prepared at the end of every quarter and shared to higher levels for their feedbacks to strengthen the existing surveillance system under the program.

## Surveillance procedure

Sentinel surveillance used various approaches for collecting information about all types of kala-azar patients. In first year of the study, usual passive case reporting method was adopted in all sentinel sites and patient treatment card was used for collecting information. Site supervisors prepared line listing of kala-azar cases by village/by areas for each month at the end of first year. Thus clusters of high incidence villages/areas of kala-azar were identified on the basis of the line listing and finally hot spots for kala-azar were recorded and documented.

## Reporting method

The site supervisor prepared a monthly consolidated report in the pre designed format for each kala-azar case. A one-page case report extracted from patient treatment card/other records were generated to see the consistency and completeness of reporting.

Line listing of every kala-azar, PKDL and co-infection cases were maintained at all the sentinel sites. Reporting was done through line listing process to supplement the existing reporting system.

## Statistical analysis

Descriptive statistics of the data were calculated. Proportions of patients and demographic characteristics were appropriately prescribed. Incidence rates of the disease were calculated per10, 000 inhabitants. The chi-square and wilcoxon tests were performed to test the statistical significance of differences in a classification system. The Bland-Altman plot (Bland and Altman, 1999) was adopted to compare two measurements techniques. One-sampled t-test was used based on logarithmic transformation to measure the significant differences of responses of drugs used for kala-azar treatment. Freidman test (Conover, 1999) was used to assess the

difference between responses of different anti-kala-azar drugs. The signed rank sum test was performed to measure the symmetrical differences between the sample median and the test value. All the analysis were performed at <0.05 significance level.

## Results

A total of 20968 symptomatic cases (fever more than 15 days) were screened and out of that, 2996 confirmed cases of KA and 53 cases of PKDL, total (3049) were registered from 2011 to 2014 in all the sentinel sites. Rest symptomatic cases were negative by rk-39 and as well as bone marrow/splenic aspiration (KA)/slit skin smear examination (PKDL). These cases were referred to other departments (Malaria, TB, Filaria, leprosy, viral disease, etc) of the health facilities, where sentinel sites were made for further management of their illness.

The percentage share of male and female was 56.4% and 43.6% respectively. Sentinel sites covered a total of 889 endemic villages. Out of 889 villages, statistically significant difference (p<0.001) was observed for villages reported 1–2 kala-azar cases (74%) per year when compared to villages that reported 3–5 KA cases (15%). Overall, 4% of total villages (approximately 36 villages) were reported more than 10 KA cases every year.

The overall proportion of kala-azar cases were 14.29% in the screened cases. The Baniyapur (Saran) site contributed highest proportion (35.78%), while lowest (8.67%) by Kalyanpur site (East Champaran). The proportion of new cases was 86.1% amongst all cases registered under the sentinel sites, while 13.9% of cases were old cases. Amongst old cases, 4% cases were unresponsive and 9% were relapsed. Out of the total KA cases (2996), majority (47%; 1408) were from age group 16–44 years, followed by 35% and 18% from age groups ≤ 15 years and ≥ 45 years respectively.

Overall, decreasing trend of incidence per 10 000 population was noticed from 2011 to 2014 for all sites. Kala-azar was significantly reduced by more than 50% from 4.13/10 000 to 1.75/10 000 with mean 2.61 and 95% CI (2.26–3.41) for the study period. Highest mean incidence (4.71/10000) was recorded for Madhuban PHC (East Champaran) with 95% CI as 3.77–6.88. In 2014, about (38%) sites reached to elimination target (<1case per 10000 populations). However, Baniyapur site (Saran) showed alarming situation (4.11 per 10000) in 2014 as compared to 2011 (5.63) Table 1.

Monthly distributions of disease showed higher (11.2%, 11.7%, 10.0% and 8.5%) incidences for the months of January, March, July and August respectively for every year. However, it was lesser (5.8%, 6.1%, 6.7%) for the month of April, September and November.

There was significant increase (0.08, 0.10 per 10 000 population) in PKDL and co-infection respectively in 2014 when compared to 2011 (0.03, 0.06 per 10 000 population) (Fig 2).

Analysis of district-wise trends of PKDL, showed overall rising trend in almost all the districts except East Champaran for the study period. A sharp increase in number of PKDL cases was observed for Saran district in 2014 when compared to 2013. The numbers of PKDL cases were almost double in Muzaffarpur and Samastipur district in the year 2013 when compared to 2011. On the whole, co-infections were also found almost double (29) in 2014 when compared to 2011 (16), and this difference was statistically significant (p<0.05), which showed rising trends in cases of co-infections. In East Champaran district, co-infection was found very alarming i.e. more than 200% times increased in number of co-infection cases from 2011 to 2014. The overall proportion of co-infections and PKDL were 2.80% and 1.74% respectively. The demographic distribution of overall illness revealed that overall; VL plus co-infection were slightly higher (55.6%) for males as compared to females (44.4%). Similarly, proportion of PKDL was also higher (52.8%) for males as compared to females (47.2%). The prevalence of co-infection was recorded highest (3.34%) for the age-group 16–44 years. Majority of the cases

**Table 1. Sentinel site-wise kala-azar incidence/10,000 population.**

| Sentinel sites | 2011 | 2012 | 2013 | 2014 | Mean | Median | *CI* |
|---|---|---|---|---|---|---|---|
| **East Champaran** | | | | | | | |
| District Hospital | 3.53 | 2.45 | 1.27 | 0.96 | 2.05 | 1.86 | 1.70–2.87 |
| Kalyanpur PHC | 3.10 | 3.31 | 1.73 | 1.41 | 2.39 | 2.42 | 2.10–3.06 |
| Madhuban PHC | 9.01 | 4.82 | 3.16 | 1.86 | 4.71 | 3.99 | 3.77–6.88 |
| **Muzaffarpur** | | | | | | | |
| Medical College (SKMCH) | 1.15 | 2.14 | 0.87 | 2.40 | 1.64 | 1.65 | 1.42–2.16 |
| District Hospital | 2.31 | 6.86 | 2.90 | 1.60 | 3.42 | 2.61 | 2.71–5.06 |
| Paroo PHC | 6.77 | 2.76 | 2.52 | 2.38 | 3.61 | 2.64 | 2.97–5.08 |
| Sahibganj PHC | 5.55 | 3.26 | 2.26 | 1.01 | 3.02 | 2.76 | 2.44–4.36 |
| **Samastipur** | | | | | | | |
| District Hospital | 0.35 | 0.21 | 0.10 | 0.85 | 0.38 | 0.28 | 0.28–0.61 |
| Dalsinghsarai PHC | 8.49 | 3.85 | 1.34 | 1.76 | 3.86 | 2.81 | 2.87–6.15 |
| Shahpur Patori PHC | 2.78 | 1.46 | 0.74 | 0.93 | 1.48 | 1.20 | 1.20–2.12 |
| **Saran** | | | | | | | |
| Sadar Hospital, Saran | 1.17 | 1.87 | 1.16 | 1.05 | 1.31 | 1.17 | 1.20–1.57 |
| Baniyapur PHC | 5.63 | 2.68 | 2.74 | 4.11 | 3.79 | 3.43 | 3.37–4.76 |
| Parsa PHC | 6.29 | 4.27 | 1.23 | 2.47 | 3.57 | 3.37 | 2.90–5.10 |
| **Total** | **4.13** | **2.88** | **1.67** | **1.75** | **2.61** | **2.32** | **2.26–3.41** |

(87%) were from rural areas. In total, 84 cases were identified as with concurrent illness (co-infection) and proportion of kala-azar cases with concurrent illness were 2.80%. Out of that, majority (56%) were suffering from HIV-VL co-infection, followed by (13%) as TBVL co-infections. The liver disease also contributed significantly (8%). The proportion of HIV-VL co-infections was significantly higher (1.6%; p<0.05) as compared to other co-infections. As regard to fever history of kala-azar cases, significantly higher proportion (60.6%) was with fever between 15–30 days, followed by (32.5%) with 30–45 days of fever. Analysis further revealed that, the patients having fever of longer duration (> 30 days) was with significant weight loss. Significantly higher proportions (61%) of kala-azar cases were with 2–3 cm spleen size when compared with 3–4 cm (about 18%) at the time of diagnosis. This difference was found statistically significant (p<0.05). However, larger spleen (4–6 cm) size was also noticed for only negligible proportion. Out of the female patients (1329), about 47% were from reproductive age-group. Almost, 50% of the reproductive age group patients were from most sensitive child bearing age group (21–30 yrs). Amongst the reproductive age-group patients (625), 3.7% (23) were found pregnant. None of the pregnant woman encountered with teratogenic effect in the delivered child.

## Demographic characteristics of the sentinel sites

Analysis further revealed that maximum cases (47%) were in the age group of 16–44 years followed by (35%) in the age group of ≤ 15 years. Proportions of male in all age groups were higher and found statistically significant (Chi-square test = 7.6; *P = 0.026*). Percentage of kala-azar cases was found in decreasing order right from secondary level to post graduation level education. Results revealed that significantly higher proportion (69%) of kala-azar cases were from up to middle level of education, and out of that, 33% were illiterate ones. Significantly lower proportion (only 9%) of kala-azar cases were from graduation and above level of education status. The educational status between male and female categories were also found

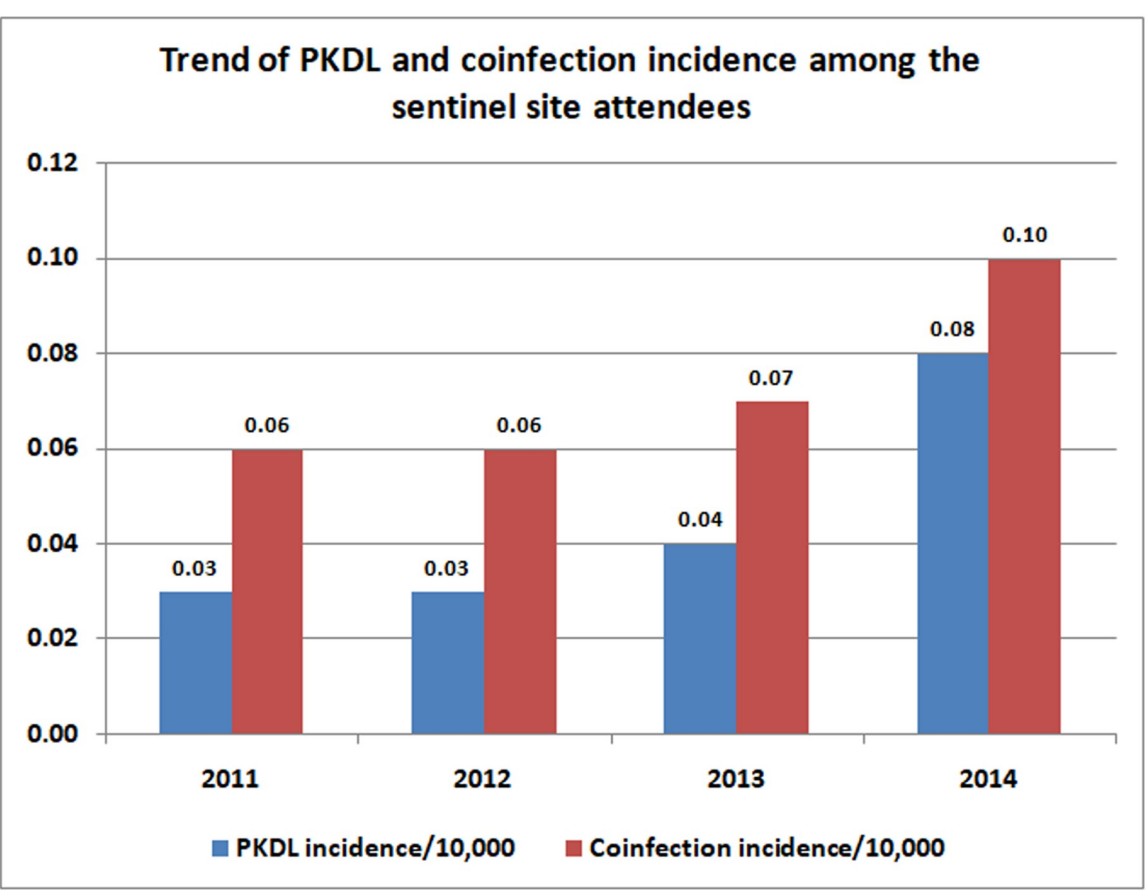

**Fig 2. Trends of PKDL and co-infection incidence among the sentinel site attendees.**

statistically significant (Wilcoxon test; P = *0.007*). About, 48% of kala-azar cases were from low socio-economic (agricultural laborer & laborer) groups. A very negligible proportion (9%) of kala-azar cases were from upper socio-economic groups. Good proportions (7%) of cases were from children below 6 years and aged individuals. The male and female categories among the different occupational groups showed statistically significant (*P<0.001*) result. Overall, significantly higher proportion (51%) of kala-azar cases were from schedule caste categories, followed by (33%) from other backward caste (OBC) categories as compared to general caste categories (16%). Social category between male and female groups were not found statistically significant (*P = 0.77*). Table 2.

The Bland-Altman plot was used to measure the relationship between the differences and the averages of male and female group to look for any systematic biases and to demarcate any possible outliers between demographic characteristics in the sentinel sites. (Fig 3).

### Diagnostic analysis

A total of 20,968 potentially suspected individuals, comprising 11285 males and 9683 females were screened for kala-azar with rK-39 rapid strip test. Out of that, 3049 (56.4% males and 43.6% females) were found rK-39 positive. The overall, percentage of rK-39 positivity was about 14%, consistent in males and females. The rK–39 positivity was slightly higher in the

**Table 2.  Demographic characteristics of kala-azar cases by sites (N = 2996, Male = 1667; Female = 1329).**

| Variable | Category | Male | | Female | | Total | |
|---|---|---|---|---|---|---|---|
| | | (N) | (%) | (N) | (%) | (N) | (%) |
| **Age** | ≤ 15 | 560 | 33.6 | 489 | 36.8 | 1049 | 35.0 |
| | 16–44 | 781 | 46.9 | 627 | 47.2 | 1408 | 47.0 |
| | ≥ 45 | 326 | 19.6 | 213 | 16.0 | 539 | 18.0 |
| **Educational status** | Illiterate | 553 | 33.2 | 434 | 32.7 | 987 | 32.9 |
| | Literate | 119 | 7.1 | 108 | 8.1 | 227 | 7.6 |
| | Primary | 292 | 17.5 | 264 | 19.9 | 556 | 18.6 |
| | Middle | 176 | 10.6 | 116 | 8.7 | 292 | 9.8 |
| | Secondary | 214 | 12.8 | 176 | 13.2 | 390 | 13.0 |
| | Higher Secondary | 137 | 8.2 | 129 | 9.7 | 266 | 8.9 |
| | Graduation | 109 | 6.5 | 79 | 5.9 | 188 | 6.3 |
| | Post-Graduation & above | 67 | 4.0 | 23 | 1.7 | 90 | 3.0 |
| **Occupational Status** | Farmer | 176 | 10.6 | 92 | 6.9 | 268 | 9.0 |
| | Agricultural Labor | 455 | 27.3 | 363 | 27.3 | 818 | 27.3 |
| | Laborer | 458 | 27.5 | 156 | 11.7 | 614 | 20.5 |
| | Business | 94 | 5.6 | 23 | 1.7 | 117 | 3.9 |
| | Service | 62 | 3.7 | 95 | 7.2 | 157 | 5.2 |
| | Students | 304 | 18.2 | 232 | 17.5 | 536 | 17.9 |
| | Housewife | 0 | 0.0 | 281 | 21.1 | 281 | 9.4 |
| | None | 118 | 7.1 | 87 | 6.6 | 205 | 6.8 |
| **Social Category** | General | 277 | 57.9 | 201 | 42.1 | 478 | 15.9 |
| | OBC | 566 | 56.4 | 437 | 43.6 | 1003 | 33.5 |
| | SC | 824 | 54.4 | 691 | 45.6 | 1515 | 50.6 |

age-group 16–44 years for both the sexes. A total of 608 cases were subjected to different tests to rule out relapse, re-infection, co-infection and PKDL cases. The sensitivity of splenic aspirate was found significantly higher (94.3%), followed by bone marrow (80.1%). Altogether, 53 (1.7%) cases were confirmed as PKDL with the help of slit-skin smear /skin-biopsy. However, the sensitivity of slit-skin smear / skin-biopsy was found 55.2%. As regard to co-infection, maximum sensitivity was recorded as 42.7% for ELISA (HIV). Utilization of laboratory services was greatly improved Table 3.

### Response of different anti-kala-azar drugs administered at sentinel sites

The higher proportion of kala-azar cases were treated with miltefosine (43.9%) as compared to other drugs viz. amphotericin-B (11.8%), Miltefosine + Paramomycin (5.1%), SAG + Miltefosine (4.8%) and AmBisome (2.5%). Good proportion of kala-azar cases (27.0%) were also treated with SAG. Statistically significant differences in responses were observed for SAG (t-test = 6.98; *P<0.0009*), SAG + Miltefosine (t-test = 7.49; *P<0.0007*), Miltefosine (t-test = 5.57; *P<0.002*), Amphotericin B (t-test = 4.46; *P<0.002*), Miltefosine + Paramomycin (t-test = 7.93; *P<0.03*) and AmBisome (t-test = 2.06; *P<0.09*).

Friedman test showed statistically significant difference between responses of different anti Kala-azar drugs (F = 25.0, P<0.004). The initial and final cure rate of AmBisome was found excellent (100%). Whereas, other drugs viz Miltefosine + Paramomycin (combination therapy) had also shown comparable responses (100%; initial & 99.3%; final cure) with that of AmBisome. Similarly, the initial and final cure rate of Amphotericin B was also found significantly

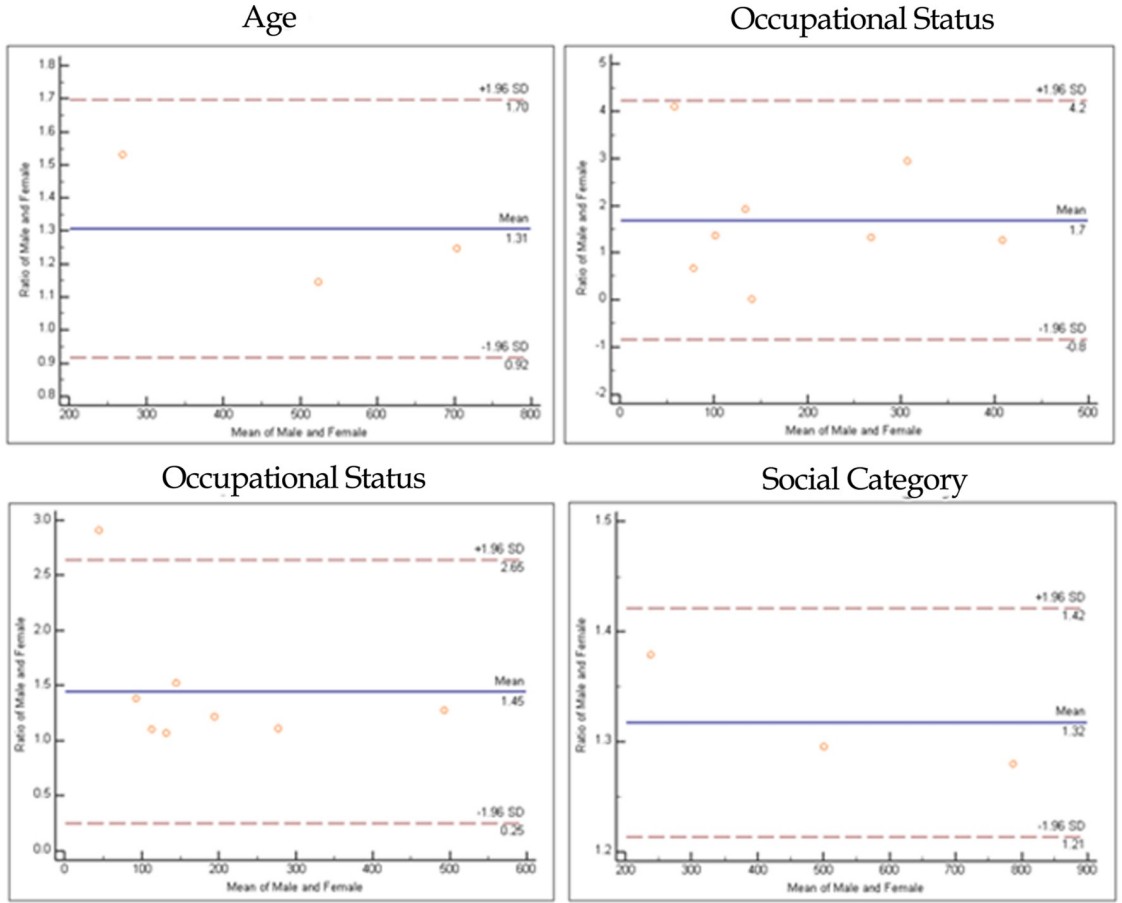

**Fig 3. Bland-Altman plot of demographic characteristics of the sentinel sites.**

higher (99.7% & 98% respectively). As far as miltefosine is concerned, final cure rate was also promising (94.1%). The response of SAG was found very poor in the treatment of kala-azar and a big difference was observed in the initial (88.1%) and final cure rate of SAG (59.3%). Table 4.

**Table 3. Distribution of other diagnostic tests for co-infection, PKDL, relapse and re-infection.**

| Categories | Male | | Female | | Total | |
|---|---|---|---|---|---|---|
| | # Tested | Found positive n (%) | # Tested | Found positive n (%) | # Tested | Found positive n (%) |
| ELISA (HIV) | 63 | 29 (46) | 47 | 18 (38.3) | 110 | 47 (42.7) |
| BM | 93 | 74 (79.6) | 58 | 47 (81) | 151 | 121 (80.1) |
| SPA | 96 | 89 (92.7) | 61 | 59 (96.7) | 157 | 148 (94.3) |
| Skin Smear | 50 | 28 (56) | 46 | 25 (54.3) | 96 | 53 (55.2) |
| Microscopy/RDT for Malaria | 8 | 2 (25) | 5 | 1 (20) | 13 | 3 (23.1) |
| Sputum for AFB/RDT for TB | 21 | 8 (38.1) | 14 | 3 (21.4) | 35 | 11 (31.4) |
| Other (Hematogram & Biochemistry) | 22 | 9 (40.9) | 24 | 14 (58.3) | 46 | 23 (50) |

#:Number, ELISA:Enzyme-Linked Immunosorbent Assay, HIV:Human Immunodefieciency Virus, BM:Bone Marrow, SPA:Splenic Aspiration, RDT for Malaria:Rapid Diagnostic Kit for Malaria, AFB:Acid-Fast Bacillus, RDT for TB:Rapid Diagnostic Kit for TB & TB:Tuberculosis

**Table 4. Response of drugs administered for the treatment (N = 2849).**

| Description of drugs | No. treated | ICR n(%) | FCR n(%) | URR n(%) | Relapse Rate n(%) | LAMA/ Defaulter Rate n(%) | Death Rate n(%) |
|---|---|---|---|---|---|---|---|
| SAG | 808 | 655 (81.1) | 479 (59.3) | 153 (18.9) | 176 (21.8) | 8 (1.0) | 10 (1.2) |
| SAG + Miltefosine | 145 | 142 (97.9) | 126 (86.9) | 3 (2.1) | 16 (11.0) | 15 (10.3) | 2 (1.4) |
| Miltefosine | 1314 | 1306 99.4) | 1236 (94.1) | 8 (0.6) | 70 (5.3) | 43 (3.3) | 12 (0.9) |
| Amphotericin B | 354 | 353 (99.7) | 347 (98.0) | 1 (0.3) | 6 (1.7) | 7 (2.0) | 2 (0.6) |
| Miltefosine + Paramomycin | 153 | 153 (100.0) | 152 (99.3) | 0 (0.0) | 1 (0.7) | 0 (0.0) | 1 (0.7) |
| AmBisome | 75 | 75 (100.0) | 75 (100.0) | 0 (0.0) | 0 (0.0) | 0 (0.0) | 0 (0.0) |

LAMA = 73: Treated at RMRI &MSF = 127) ICR: Initial Cure Rate, FCR: Final Cure Rate, URR: Unresponsiveness Rate

The rate of un-responsiveness of SAG was found significantly higher (18.9%) as compared to rest of the drugs, whereas un-responsiveness rate for SAG + Miltefosine (combination) was found negligible (2.1%). The results of the signed rank sum test showed significant symmetry of unresponsiveness rate ($P<0.03$). Similarly, relapse rate of SAG was also found significantly higher as compared to other drugs (95%CI 0.2165 to 19.7035; $P<0.03$). There was significant differences of the LAMA/defaulter rate between the anti-kala-azar drugs (95%CI -1.2634 to 6.8634; $P<0.03$). However, no significant differences of death rate were observed between the different anti-kala-azar drugs ($P<0.06$).

## Adverse Drug Reactions (ADR)

Besides, baseline clinical examination (0 day), the study subjects were clinically examined on each visit to assess clinical improvement as well as adverse events (AE), if any. Altogether, 401 AE's were reported in the study subjects, of which gastrointestinal side effects, such as nausea (26.4%), diarrhea (22.2%), abdominal pain (12.7%), and dehydration (9.7%), were the most common recorded complaints. All patients were graded under CTC; 256 (64%) were in grade I, 75 (19%), 52 (13%), 7 (1.5%), and 11 (2.5%) were in grades II, III, IV, and V, respectively.

## Severe ADR

Of 11 deaths reported as serious adverse events (SAE's), the explored cause of deaths were extreme diarrhea (N = 7), kala-azar-pulmonary tuberculosis co-infection (N = 2), HIV/AIDS co-infection (N = 1), and acute pancreatitis (N = 1). A total of 33 patients required hospitalization during domiciliary mode of treatment with miltefosine to manage the AE's. In those patients having ADR, 6% were suffering from severe ADR's, 58% were suffering from moderate ADR's, whereas the rest were suffering from mild ADR's.

## Focused villages of sentinel sites having > 10 cases/year during the study period

Out of 889 endemic villages, only 6 (0.7%) villages under two sentinel sites viz. Kalyanpur and Madhuban of East Champaran district contributed 76 kala-azar cases in every year and thus a total of 304 cases were from these villages. Out of the total cases registered (2996) at sentinel sites, 10% share were from these six highly endemic villages. The highest cases (19) were

**Table 5. Concordance between sentinel surveillance and KMIS data by district.**

| Sl. | District | 2011–2014 | |
|---|---|---|---|
| | | # of cases detected by SS | # (%) of cases captured by KMIS |
| 1 | East Champaran | 742 | 725 (97.7) |
| 2 | Muzaffarpur | 1042 | 1008 (96.7) |
| 3 | Samastipur | 445 | 429 (96.4) |
| 4 | Saran | 767 | 688 (89.7) |
| **Total** | | **2996** | **2850 (95.1)** |

SS: Sentinel surveillance, KMIS: Kala-azar management information system

recorded in Shambu Chak village under Kalyanpur sentinel site of East Champaran district in each year.

## Concordance between KMIS and sentinel sites data

Analysis primarily consists of determining the discordance between the two sources. The cases registered under sentinel sites were compared to that of VL cases captured under existing surveillance system and it may provide preliminary leads to efforts required to strengthen KMIS. Analysis showed that significantly higher proportions (95%) of cases were captured under KMIS. KMIS data was almost close to actual disease burden. The similar pattern was noted for every district except Saran district (89.7%) Table 5.

## Discussion

The Government of India has taken all possible efforts to eliminate kala-azar by the year 2020 but still the incidence of the diseases continues to be high in 64 affected blocks in the country, and even the disease is spreading in newer areas. Good quality kala-azar surveillance in India always poses challenge to the programme. For effective control of VL disease within the targeted time period, a proper documentation of the extent of problem in terms of kala-azar, PKDL, co-infections badly required. Proper monitoring of kala-azar elimination goals (<1 case/ 10 000 population) and deaths are warranted. The health facility-based sentinel site surveillance may be useful to observe trends of kala-azar morbidity. Further, it may also be useful in improving case management. The basic purpose of this system was to detect KA, PKDL outbreaks through regular analysis and reporting to plan appropriate public health responses. It is always challenging to capture complete and accurate patient level data in the existing surveillance system. The use of diagnostic testing may improve patient care and reduce the use of anti KA drugs unnecessary. The sentinel site surveillance may also provide information on treatment outcomes. This approach was attempted first time in Bihar under research mode with fewer additional resources by including a few facilities at different levels to provide reliable and complete information on kala-azar, PKDL and co-infections. We found that throughout four years of sentinel surveillance, kala-azar was an important contributor to both inpatient and outpatient.

The overall proportion of kala-azar cases were 14.29%, amongst the symptomatic cases (20968; fever more than 15 days) attended sentinel sites. This information is not available for routine existing surveillance system (Program data). However, a study conducted in India Nepal and Bangladesh reported proportion of kala-azar cases as 2.6% amongst symptomatic in India [14]. The proportion of new and old cases (unresponsive and relapse cases) were 86.1% and 13.9% respectively. Thus, 13.9% of old VL cases were also diagnosed and treated at these

sentinel sites. On the other hand, generally treatment of unresponsive and relapse cases were not done at the existing health facilities under the program. Such cases mostly refer to higher health facilities in the existing health system. Overall, decreasing trends of incidence per 10 000 population was noticed from 2011 to 2014 for all sites. Kala-azar was significantly reduced by more than 50% from 4.13/10 000 to 1.75/10 000 with mean 2.61 and 95% CI (2.26–3.41) for the study period. In 2014, about (38%) sites reached to elimination target (<1case per 10000 populations). The results are identical as described in Accelerated Plan for Kala-azar Elimination, 2017 [15]. Suman Rijal et al. documented that, kala-azar cases have declined consistently from over 77 000 in the year1992 to less than 7000 cases in the year 2016. About, 8% of endemic units were above the threshold at the end of 2017 [16]. Another study reported that VL incidence (cases/10,000/year) was reduced from 12.3 in the year 2007) to 0.9 in the year 2015. This is just below the target of World Health Organization's threshold as a public health problem [17]. Our study revealed that, there was no uniform pattern of trends for medical college level sites. The possible explanation for the trend obtained for medical college level may be, training, frequent stakeholders meeting, availability of free diagnostics and single dose AmBisome treatment and payment of various incentives under programme.

Monthly distributions of disease showed higher (11.2%, 11.7%, 10.0% and 8.5%) incidences for the months of January, March, July and August respectively for every year. However, it was lesser (5.8%, 6.1%, 6.7%) for the month of April, September and November. A study conducted by Paritosh Malviya et al. also reported highest peak in March–April and another one in July [18]. Whereas a study conducted in Sri Lanka reported seasonal trend of leishmaniasis incidence with a peak in July to September between the two monsoon periods and a drop in the number of reported cases compared to rest of the months [19]. The possible explanations of seasonal variation for higher incidence of kala-azar in our study were presence of sand flies in abundance in endemic areas in rainy season (favorable humidity & temperature), which include highest number of gravid sand flies and on the other side, more active kala-azar cases are also found during January, March, July and August in the endemic areas. Therefore, there is chance of higher number of parasites in sand flies while taking a blood meal from an infected human host, present in the endemic areas. It has also been documented in the literatures that pre and post monsoon season (February to August) sand fly resurgence is much higher as compared to other seasons. Further, shorter incubation period (1–2 months) increased the disease transmission since February to August. Consequently, the infectivity of sand flies may be higher during these months.

The number of PKDL cases was almost double in Muzaffarpur and Samastipur district in the year 2013 when compared to 2011. The possible explanations for this may be the majority of PKDL (about 80%) cases are with previous history of VL treatment. Traditionally, Muzaffarpur is one of the highest endemic districts since long time. In India, PKDL usually occurs at an interval of 2–3 years in 5–10% of cases after successful treatment of VL. These two districts had history of VL treatment with SAG. Previously (since 2008–2011) most of the VL cases had been treated with SAG. Inadequate and inappropriate treatments are documented in the literatures as associated with the development of PKDL after VL treatment. Development of PKDL in about 10% of VL cases, who had an interruption in their treatment, has also been present in the literatures. The common reasons for incomplete treatment were concern about loss of daily wages, loss of school days and do not feel sick. The risk of developing PKDL in our study areas seems to be associated with SAG treatment (majority of VL cases treated with SAG) and incomplete SAG treatment. PKDL development is also found in VL patients treated with other anti-leishmanial drugs (e.g. miltefosine, amphotericin B and paromomycin) in our study areas.

On the whole co-infection was also found almost double (29) in 2014 when compared to 2011 (16), which showed rising trends for cases of co-infections too. In East Champaran district, co-infection was found very alarming i.e. more than 200% times increased in number of co-infection cases from 2011 to 2014. The overall proportion of co-infections and PKDL were 2.80% and 1.74% respectively. Finding may be vital for programme perspective and other stakeholders. It was observed that, the incidence of kala-azar has decreased while incidences of PKDL and co-infections have been increased in a span of four years of the study.

Das et al. documented that, the prevalence of PKDL was high (9.7 per 10 000) [20]. However, Singh RP et al. reported the prevalence of confirmed PKDL as 4.4 per 10 000 individuals [21]. On the other hand, V Ramesh et al. found an upward trend in reporting of PKDL cases to Safdarjung Hospital, New Delhi, India since the year 2005 [22]. The demographic distribution of overall illness revealed that, overall VL+co-infection were slightly higher (55.6%) for males as compared to females (44.4%). Another researcher Sakib Burza et al reported the relative risk (RR) of being HIV-positive was 3.7 times higher for males [23]. Similarly, proportion of PKDL was also higher (52.8%) for males as compared to females (47.2%). This finding was identical to that of a study conducted by V Ramesh et al. [22] The prevalence of co-infection was recorded highest (3.34%) for the age-group 16–44 years as reported by Sakib Burza et al. [23] Majority of the cases (87%) were from rural areas. The possible explanations may be prevailing situation of the rural areas viz. favorable environmental conditions for vector, poor level of literacy, lack of awareness, low socio-economic conditions, poor living standard, and vegetation around houses and presence of cattle-shed inside houses as compared to urban areas. In total, 84 cases were identified as with concurrent illness (co-infection) and proportion of kala-azar cases with concurrent illness was 2.80%. Out of that, majority (56%) were suffering from HIV-VL co-infection, followed by (13%) TB-VL co-infection. Liver disease also contributed significantly (8%). The proportion of HIV-VL co-infection was significantly higher (1.6%; p<0.05) as compared to other co-infection. This result was in line with other earlier studies conducted elsewhere [24–26]. As regard to fever history of kala-azar cases, significantly higher proportion (60.6%) was with fever history between 15–30 days. There was significant weight loss in the kala-azar patients who have fever of longer duration (> 30 days). Significantly higher proportions (61%) of kala-azar cases were with 2–3 cm spleen size when compared with 3–4 cm (about 18%) at the time of diagnosis. This difference was found statistically significant (p<0.05). Similar findings were obtained by earlier studies conducted elsewhere [27–32]. However, large spleen (4–6 cm) size was also noticed for only negligible proportion under this study. Out of the female patients (1329), about 47% were from reproductive age-group. Almost, 50% of the reproductive age group patients were from most sensitive child bearing age group (21–30 yrs). Amongst the reproductive age-group patients (625), 3.7% (23) were found pregnant. None of the pregnant woman encountered with teratogenic effects in the delivered child. Identical result was obtained by a study conducted in Bihar by Pandey K et al. [33] Maximum cases (47%) were in the age group of 16–44 years and proportions of male in all age groups were higher and found statistically significant (Chi-square test = 7.6; *P<0.026*). These findings were coinciding with other earlier conducted studies [34, 35]. Our results further showed that, significantly higher proportion (69%) of kala-azar cases were from up to middle level of education, and out of that, 33% were illiterate ones. Significantly lower proportion (only 9%) of kala-azar cases were from graduation and above level of education status. The educational status between male and female categories were also found statistically significant (Wilcoxon test; *<0.007*). About, 48% of kala-azar cases were from low socio-economic (agricultural laborer) groups. A very negligible proportion (9%) of kala-azar cases were from upper socio-economic groups. Good proportion

(7%) of cases was from children below 6 years and aged individuals. The male and female categories among the different occupational groups showed statistically significant (*P<0.001*) result. Significantly higher proportion (51%) of kala-azar cases was from schedule caste categories, which was similar with other studies conducted in other parts of world [27, 36]. The possible reasons for this may be mainly due to socio-economic conditions and not due to genetic implications.

## Diagnostics analysis

A total of 20,968 potentially suspected individuals, comprising 11285 males and 9683 females were screened for kala-azar with rK-39 rapid strip test. Out of that, 3049 (56.4% males and 43.6% females) were found rK-39 positive. The overall, percentage of rK-39 positivity was about 14%, consistent in males and females. The rK–39 positivity was slightly higher in the age-group 16–44 years for both the sexes. A total of 608 cases were subjected to different tests to rule out relapse, re-infection, co-infection and PKDL cases. The sensitivity of splenic aspirate was found significantly higher (94.3%), followed by bone marrow (80.1%). However, the sensitivity of slit-skin smear/ skin-biopsy was found 55.2%. As regard to co-infection, maximum sensitivity was recorded as 42.7% for ELISA (HIV). These results were similar with that of other studies conducted in different parts of world [37–39].

## Response of different anti-kala-azar drugs administered at sentinel sites

The higher proportion of kala-azar cases were treated with miltefosine (43.9%) as compared to other drugs viz. amphotericin-B (11.8%), Miltefosine+Paramomycin (5.1%), SAG + Miltefosine (4.8%) and AmBisome (2.5%). Good proportion of kala-azar cases (27.0%) were also treated with SAG. Statistically significant differences in responses were observed for SAG (t-test = 6.98; *P<0.0009*), SAG + Miltefosine (t-test = 7.49; *P<0.0007*), Miltefosine (t-test = 5.57; *P<0.002*), Amphotericin B (t-test = 4.46; *P<0.002*), Miltefosine + Paramomycin (t-test = 7.93; *P<0.03*) and singe dose AmBisome (t-test = 2.06; *P<0.09*). The initial and final cure rate of AmBisome was found excellent (100%). Possible explanation may be the impact of elimination initiative (enhance awareness among people) highest proportion comes to the sites by own. Whereas, other drugs viz Miltefosine + Paramomycin (combination therapy) had also shown comparable response (100%; initial & 99.3%; final cure) with that of AmBisome. Similarly, the initial and final cure rate of Amphotericin B was also found significantly higher (99.7% & 98%) respectively. As far as miltefosine is concerned, final cure rate was also promising (94.1%). The response of SAG was found very poor in the treatment of kala-azar and a big difference was observed in the initial (88.1%) and final cure rate of SAG (59.3%). The response of SAG was found very poor. Findings were identical with other study conducted by Om Prakash Singh et al in Bihar, India [40]. The rate of un-responsiveness of SAG was found significantly higher (18.9%) as compared to rest of the drugs, while un-responsiveness rate for SAG + Miltefosine (combination) was found as negligible (2.1%). The results of the signed rank sum test showed significant symmetry of unresponsiveness rate (*P<0.03*). The relapse rate of SAG was found significantly higher as compared to other drugs (95%CI 0.2165 to 19.7035; *P<0.03*). Interestingly, no relapse was observed for AmBisome and Miltefosine + Paramomycin, however, it was negligible (<1%) for Miltefosine alone and Amphotericin B. These were in the line of findings of other studies conducted in Indian sub-continent [41–43]. There was not a single case of LAMA/defaulter with AmBisome and Miltefosine + Paramomycin. However, no significant differences of death rate were observed between the different anti-kala-azar drugs (*P>0.06*). These findings were similar as reported by Sakib Burza et al [43].

### Adverse Drug Reaction (ADR)

Altogether, 401 AE's were reported in the study subjects, of which gastrointestinal side effects, such as nausea (26.4%), diarrhea (22.2%), abdominal pain (12.7%), and dehydration (9.7%), were the most common recorded complaints. All patients were graded under CTC; maximum (64%; 256) were in grade-I, followed by (19%; 75) in grade-II.

### Severe ADR

Of 11 deaths reported as Serious Adverse Event (SAE's), the explored cause of deaths were extreme diarrhea (N = 7), kala-azar-pulmonary tuberculosis co-infection (N = 2), HIV/AIDS co-infection (N = 1), and acute pancreatitis (N = 1).

### Focused villages of sentinel sites having > 10 cases/year during the study period

Out of the total cases registered (2996) at sentinel sites, 10% share were from only 6 villages (304 cases) under two sentinel sites Kalyanpur and Madhuban of East Champaran district. The highest cases (19) were recorded in Shambu Chak village under Kalyanpur sentinel site of East Champaran district in each year. These villages are crucial for the program manager and policy makers and focused intervention need to be initiated on urgent basis to meet the target of kala-azar elimination. The possible explanations may be the continued transmission of the disease due to favorable conditions to sand fly in these pockets and lacking of various program interventions. Villages having more than 10 kala-azar cases per year required immediate specific intervention to control the disease and to check the transmission cycle. A researcher, Canjun Zheng et al. found similar results. He has concluded that VL continues to be a serious public health problem in Kashi Prefecture, China. It is a relatively high-risk area and there are several hot spots. The detection of spatial and spatiotemporal patterns can provide significant information for prevention and control measures [44]. However, another study conducted in India recorded the spatially clustered patterns with significant differences by village. The hotspots showed the spatial trend of kala-azar diffusion (P < 0.01) [45].

### Concordance between KMIS and Sentinel sites data

Analysis showed that significantly higher proportions (95%) of cases were captured under KMIS. KMIS data was almost close to actual disease burden. The possible explanations may be the various enforced interventions rooted under kala-azar elimination programme in endemic districts by the National Program.

Capturing complete and accurate patient level data in the existing surveillance system is always challenging. Generally treatment of unresponsive, relapse and re-infection cases were not managed at the existing health facilities under the program. Such cases mostly refer to higher health facilities (level-III facilities, which include medical colleges, kala-azar research institutions viz RMRIMS, KMRC, Balaji Utthan Sansthan, etc in the state). Our study managed 13.9% unresponsive; relapse and re-infection cases of VL at these sentinel sites. Study further captured 53 confirmed cases of PKDL with the help of slit-skin smear /skin-biopsy. The majority (98%) of patients with suspected VL; PKDL & Co-infections were managed in the sentinel sites itself. The results further revealed that incidence of PKDL and co-infections have increased in a span of four years of the study. A total of 608 cases were subjected to different tests to rule out relapse, re-infection, co-infection and PKDL cases. Interestingly, no relapse was observed for AmBisome and Miltefosine + Paramomycin, however, it was negligible

(<1%) for Miltefosine alone and Amphotericin B. The study reported grade-I (64%) and grade-II (19%) AE's and SAE are in 11 cases. These data may be vital and missing in existing kala-azar surveillance system. Thus present study showed significant improvements in kala-azar, PKDL and co-infection cases management at the sites as compared to existing public health facilities. At the onset of the sentinel site, majority of suspected kala-azar cases were diagnosed and treated at the facility itself without further referral to elsewhere. Utilization of laboratory services were much improved at the sentinel sites. The use of diagnostic testing may improve patient care and reduce the uses of unnecessary anti KA drugs. Study further identified the hot spots of VL. These villages are crucial for the program manager and policy makers and focused intervention need to be initiated on urgent basis to meet the target of kala-azar elimination. The possible explanations may be the continued transmission of the disease due to favorable conditions to sand fly in these pockets and lacking of various program interventions. Data coming out from this study may also be vital for informing routine program decision making. It may further be useful in monitoring commodity stocks and demands. Program managers may use results for addressing a variety of local needs on various aspects. This method may also be utilized in evaluating program effectiveness. The lessons learned from this study may benefit other initiatives aimed to improve kala-azar case management in other health care facilities. It may be helpful for changing the practices of health care workers in Bihar, India. Such systems may be utilized for deciding operational strategies to locate foci of increased KA, PKDL, and co-infection transmission. This system provided the basis that how fewer additional resources may make the system more sensitive/effective. It may also provide a model for improving laboratory services and KA, PKDL and co-infection case management in other health facilities in Bihar without further referral. Finally, it may be used as a cost-effective approach for capturing real-time kala-azar data in national kala-azar elimination programme.

The similar pattern was noted for every district except Saran district (89.7%). This finding was not coinciding with that of other study (2010) conducted much earlier in India. The referred study reported the overall magnitude of VL cases not reported to the government agencies was higher by a factor 4.17 (95% CI = 3.75–4.63) [46]. The possible explanations may be the various enforced interventions rooted under kala-azar elimination programme in endemic districts by the national program. Niyamat Ali Siddiqui et al documented that a small proportion (38; 15.8%) of KA cases was not present in the public health system record. This was similar to present study finding [47].

Low number of cases notification by private health care providers may not be a big concern for the programme. It means that kala-azar control programme is doing well at the moment (under the era of kala-azar elimination). However, it can be improved at private health care provider's level through frequent training/re-training for detection of VL/PKDL cases in the endemic areas, making diagnostic (rk-39) and anti KA drugs in the open market freely. Government should make provision of incentive for them. Such approach enhance the detection of hidden and under reported case of VL/PKDL.

## Conclusions

This paper discusses various difficulties coming into programme implementation. It is focusing on existing kala-azar KMIS performance: 1) timely and accurate data reporting process; 2) accounting extent of problem in terms of kala-azar, PKDL, co-infections; 3) opportunity to improve kala-azar case management; 4) to identify outbreaks and hot spot; 5) capturing complete and accurate patient-level data; and 6) monitoring progress towards the national kala-azar elimination goals.

The overall proportion of co-infection and PKDL were 2.80% and 1.74% respectively. In total, 84 cases were identified as with concurrent illness (co-infection) and proportion of kala-azar cases with concurrent illness was 2.80%. Out of that, majority (56%) were suffering from HIV-VL co-infection, followed by (13%) TB-VL co-infection. The liver disease also contributed significantly (8%). The proportion of HIV-VL co-infection was significantly higher (1.6%; p<0.05) as compared to other co-infection. A total of 608 cases were subjected to different tests to rule out relapse, re-infection, co-infection and PKDL cases. Altogether, 53 (1.7%) cases were confirmed as PKDL with the help of slit-skin smear /skin-biopsy. Thus present study showed significant improvements in kala-azar, PKDL and co-infection case management at the sentinel sites. It has impacted on the proportion of patients with suspected kala-azar referred for diagnostic testing at elsewhere. At the onset of the sentinel site, majority of suspected kala-azar cases were diagnosed and treated at the facility itself without further referral to elsewhere. Utilization of laboratory services were much improved at the study sites. The majority (98%) of patients with suspected VL; PKDL & Co-infections were managed in the sentinel sites itself. Friedman test showed statistically significant difference between responses of different anti Kala-azar drugs (F = 25.0, P<0.004). The initial and final cure rate of AmBisome was found excellent (100%). Whereas, other drugs viz Miltefosine + Paramomycin (combination therapy) had also shown comparable response (100%; initial & 99.3%; final cure) with that of AmBisome. Similarly, the initial and final cure rate of Amphotericin B was also found significantly higher (99.7% & 98% respectively). As far as miltefosine is concerned, final cure rate was also promising (94.1%). The response of SAG was found very poor in the treatment of kala-azar and a big difference was observed in the initial (88.1%) and final cure rate of SAG (59.3%).Treatment practices were also significantly improved as per the national guidelines. Several studies conducted in India reported delay in the diagnosis of suspected individuals even when these services are available. Data come out from this study may also be vital for informing routine program decision making. It may further be useful in monitoring commodity stocks and demands. The burden of kala-azar expected at public health facilities may be estimated with the help of findings of this study. Program managers may use results for addressing a variety of local needs on various aspects. This method may be utilized in evaluating program effectiveness.

The lessons learned from this study may improve kala-azar case management in other health facilities with fewer resources. This system provide the basis that how fewer additional resources may make the system more sensitive/effective. The drug and diagnostic are already available in most of the existing facilities. Laboratory personnel are also available in the health facilities. Most of the health system workers were utilized for this system. Additionally, only one site supervisor was recruited per district to monitor the activities carried out in the sentinel site. However, major concern was availability of pathologists in the existing health facilities. Mainly, specific training/re-training/orientation was provided to health facility workers under this system. This may be addressed by assigning responsibility and building strong coordination amongst existing health system workers, which may be assigned to any suitable official available to health system. Frequent Training of laboratory personnel and clinician was needed to be conducted to make them efficient in discharging their assigned duties. Training/re-training/orientation may change the practices of health care workers in Bihar, India. It also adds the dimension of collection and analysis of individual patient-related information. The magnitude of exiting health facilities problem can be assessed and an effective strategy may be evolved with the help of finding of this study. Therefore, sentinel site surveillance system may be sustained in the context of India and it would be relevant too in the light of findings of this study. The success of any program required patience, flexibility, feedback from health care workers, and continuous support from the government and funding agency. The kala-azar

surveillance program described here may not be expanded beyond the sentinel sites. However, the lessons learned from this program should benefit other initiatives aimed to improve kala-azar case management in other health care facilities. It may be useful for changing the practices of health care workers in India. Such systems may be utilized for deciding operational strategies to locate foci of increased KA, PKDL, and co-infection transmission. It provides lead to how to achieve high quality surveillance data in a resource limited setting.

## Acknowledgments

We are grateful to NVBDCP, specially Dr. A.C. Dhariwal, Director and Dr. R. K. Das Gupta, Joint Director for their valuable suggestions for this study. We would also like to thank all the participating technical, project and field staff for their support. Administrative and financial assistance provided by Mr. Rajiv Ahuja, Mr. Naresh Kumar, A.O., and Mr. Udai Kumar, Sr. A.O. (F&A) greatly acknowledged. Finally, indispensable contribution of Bihar State and District Health Authorities, Health workers, Community leaders and Local residents are hereby appreciated for timely completion of the work.

## Author Contributions

**Conceptualization:** Vidya Nand Rabi Das, Niyamat Ali Siddiqui, Krishna Pandey, Alok Ranjan, Pradeep Das.

**Data curation:** Vidya Nand Rabi Das, Niyamat Ali Siddiqui, Sanjay Kumar Sinha, Md Zahid Ansari, Alok Ranjan, Vijay Pratap Singh.

**Formal analysis:** Vidya Nand Rabi Das, Niyamat Ali Siddiqui, Krishna Pandey, Sanjay Kumar Sinha, Md Zahid Ansari, Alok Ranjan, Vijay Pratap Singh.

**Funding acquisition:** Vidya Nand Rabi Das, Niyamat Ali Siddiqui, Krishna Pandey, Alok Ranjan, Pradeep Das.

**Investigation:** Vidya Nand Rabi Das, Niyamat Ali Siddiqui, Krishna Pandey, Sanjay Kumar Sinha, Md Zahid Ansari, Roshan Kamal Topno, Chandra Sekhar Lal, Alok Ranjan, Vijay Pratap Singh.

**Methodology:** Vidya Nand Rabi Das, Niyamat Ali Siddiqui, Krishna Pandey, Alok Ranjan, Vijay Pratap Singh.

**Project administration:** Vidya Nand Rabi Das, Niyamat Ali Siddiqui, Sanjay Kumar Sinha, Alok Ranjan, Pradeep Das.

**Resources:** Vidya Nand Rabi Das, Niyamat Ali Siddiqui, Sanjay Kumar Sinha, Roshan Kamal Topno, Chandra Sekhar Lal, Alok Ranjan, Vijay Pratap Singh.

**Software:** Vidya Nand Rabi Das, Niyamat Ali Siddiqui, Alok Ranjan.

**Supervision:** Vidya Nand Rabi Das, Niyamat Ali Siddiqui, Gouri Sankar Bhunia, Krishna Pandey, Sanjay Kumar Sinha, Md Zahid Ansari, Roshan Kamal Topno, Alok Ranjan, Vijay Pratap Singh, Pradeep Das.

**Validation:** Vidya Nand Rabi Das, Niyamat Ali Siddiqui, Gouri Sankar Bhunia, Krishna Pandey, Sanjay Kumar Sinha, Md Zahid Ansari, Alok Ranjan, Pradeep Das.

**Visualization:** Vidya Nand Rabi Das, Niyamat Ali Siddiqui, Sanjay Kumar Sinha, Chandra Sekhar Lal, Alok Ranjan, Pradeep Das.

**Writing – original draft:** Niyamat Ali Siddiqui, Gouri Sankar Bhunia, Sanjay Kumar Sinha, Md Zahid Ansari, Alok Ranjan.

**Writing – review & editing:** Vidya Nand Rabi Das, Niyamat Ali Siddiqui, Krishna Pandey.

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
