## [Decision Letter · Decision Letter 0]

14 Nov 2020

Dear Dr Rabidas,

Thank you very much for submitting your manuscript "Improved Kala-azar Case Management through Implementation of Health Facility-Based Sentinel Sites Surveillance System in Bihar, India" for consideration at PLOS Neglected Tropical Diseases. As with all papers reviewed by the journal, your manuscript was reviewed by members of the editorial board and by several independent reviewers. In light of the reviews (below this email), we would like to invite the resubmission of a significantly-revised version that takes into account the reviewers' comments. 

Please also find the attached file for some comments raised by one of the reviewers as well the reviewers' comments mentioned as follows:

We cannot make any decision about publication until we have seen the revised manuscript and your response to the reviewers' comments. Your revised manuscript is also likely to be sent to reviewers for further evaluation.

Sincerely,

Vahid Yazdi-Feyzabadi, PhD

Deputy Editor

Vahid Yazdi-Feyzabadi

Deputy Editor

Please also find the attached file for some comments raised by one of the reviewers as well the reviewers' comments mentioned as follows:

Reviewer's Responses to Questions

**Key Review Criteria Required for Acceptance?**

**Methods**

-Are the objectives of the study clearly articulated with a clear testable hypothesis stated?

-Is the study design appropriate to address the stated objectives?

-Is the population clearly described and appropriate for the hypothesis being tested?

-Is the sample size sufficient to ensure adequate power to address the hypothesis being tested?

-Were correct statistical analysis used to support conclusions?

-Are there concerns about ethical or regulatory requirements being met?

Reviewer #1: - The description of the materials and methods are very explorative. Some of these have little scientific value. It would be advisable to publish the protocols on a public site and refer to this site in this manuscript. By doing so, the useful information will be more accessible by non-scientific members (i.e. decision makers)

Reviewer #2: Are the objectives of the study clearly articulated with a clear testable hypothesis stated?

Yes, partially. The objective of the efficacy of anti kala-azar drugs administered at sentinel sites was not stated among the objectives.

Is the study design appropriate to address the stated objectives?

Yes, except for the efficacy of anti kala-azar drugs 

Is the population clearly described and appropriate for the hypothesis being tested?

Yes.

-Is the sample size sufficient to ensure adequate power to address the hypothesis being tested?

Yes.

Were correct statistical analysis used to support conclusions?

Yes, except for the efficacy of anti kala-azar drugs. 

-Are there concerns about ethical or regulatory requirements being met?

Yes.

Reviewer #3: The Manuscript "Improved Kala-azar Case Management through Implementation 1 of Health Facility-Based

Sentinel Sites Surveillance System in Bihar, India" is a well planned and well implemented work related to a very important public health problem in India. The work describes kala-azar, a neglected vector-borne disease, targeted for elimination, becomes impeded by high incidence of cases in several affected rural blocks of Bihar and the spread of disease in newer areas. The present study proposed to do a proper accounting of VL, PKDL and co-infection morbidity,

mortality, diagnosis, case management, hotspot identification and monitoring the impact of elimination interventions.

The manuscript warrants publication. The study hypothesis and the design is sound using proper statistical methods.

I reckon the authors have met the ethical and regulatory requirements.

**Results**

-Does the analysis presented match the analysis plan?

-Are the results clearly and completely presented?

-Are the figures (Tables, Images) of sufficient quality for clarity?

Reviewer #1: The results section is very well elaborated. 

However some attention points have to be addressed: 

- A lot of data are presented but are not discussed. By doing so, the reader is left behind to self explore the added value of all these results.

- Figure 2: the values added in the figure for the coinfection (red line) does not correspond with the scale of the incidence in the coordinate. For instance the point for 2011 (0.06) is in the figure >0.08, the point for 2014 (0.1) is in the figure >0.16. Therefor, the red line should be lower in the figure. Alse the values of 2011 and 2012 are both 0.06, however in the figure they does not seem at equal level. 

- Table 3: the “%” deserves more explanation. If I understood, it means the percentage of positive tests on the amount of test performed for the specific disease (not for all tests performed).

- Also for this table, describe the abbreviations used. 

- Related to this, the % used in the section “Diagnostics analysis” in the discussion is very confusing for the reader. It should be described what is meant by the %. 

- Table 5: It is not clear for the reviewer how the % is calculated. Please describe. Related to this, the reviewer could not interpret this part of the result section. 

- improvements to make to some difficult to understand sentences, repetitions, lack of additional data: 

 - L432-434: "A statistical significant difference was found between villages having 1-2 cases (74%) and villages with 3-5 cases (15%). Good proportion (4%) of villages was still with more than 10 cases." 

 - L439: "About half of the VL (47%) cases were in the range-group 16-44 years." This should be compared with the description of the population. What is the fraction of the 16-44 years group within the entire population? 

 - L445: "Overall, mean value of the incidence was recorded 2.61 with 95% CI as 2.26 to 3.41. "This is a repetition of line L441 and can thus be omitted.

Reviewer #2: Does the analysis presented match the analysis plan?

Yes.

Are the results clearly and completely presented?

Yes, however there are a lot of abbreviations stated for the first time without full name which made it difficult to follow the text smoothly. I also suggest the authors to provide a table for abbreviations that have been used over the manuscript.

Reviewer #3: There come some clarifications to be made 

i) The sentence "20968 symptomatic cases attended sentinel sites, 2996 cases of KA and 53 cases of

36 PKDL were registered from 889 endemic villages" ; The proportion of new and old cases was

37 86.1% and 13.9% respectively. Out of 20,968 cases 2996 cases of KA and 53 cases of

36 PKDL were registered. Where did the rest of the symptomatic go or attended to?

ii) A note on disparity in Government Vs, Private hospital based reporting system needs to be explained. Also, the observation of high prevalence of cases in some Castes. Is socioeconomic condition and/or genetic basis implicated?

**Conclusions**

-Are the conclusions supported by the data presented?

-Are the limitations of analysis clearly described?

-Do the authors discuss how these data can be helpful to advance our understanding of the topic under study?

-Is public health relevance addressed?

Reviewer #1: The conclusion are supported by the data presented in the result section. The authors discuss the data in relation with other findings, availble at international level.

Reviewer #2: -Are the conclusions supported by the data presented?

Partially.

Are the limitations of analysis clearly described?

Yes.

-Do the authors discuss how these data can be helpful to advance our understanding of the topic under study?

Yes.

-Is public health relevance addressed?

Yes.

Reviewer #3: The sentence "However, generalizability of this sentinel surveillance finding to other context remains a major

limitation of this study" needs clarification.

The manuscript highlights that the data coming out from this study may be vital for informing routine program decision making. It may further be useful in monitoring commodity stocks and demands. The burden of kala-azar expected at public health facilities may be estimated with the help of findings of this study. Program Managers may use results for

 addressing a variety of local needs on various aspects. This method may be utilized in evaluating program effectiveness.

**Editorial and Data Presentation Modifications?**

Reviewer #1: - Multiple typing errors were encountered in this manuscript. It would be useful to recheck the document by a spelling checker or re-read the final document. 

- At different positions in the text, abbreviations are used without definition. It would be advisable to define the abbreviation when first used.

See attachment for more details

Reviewer #2: (No Response)

Reviewer #3: Minor Revision that incorporates clarifications of some sentences is needed thane can be Accepted for Publication.

**Summary and General Comments**

Reviewer #1: This maunscript describes a global project for the neglected disease VL facing as well the surveillance, testing capacity, treatment and educational level of healthcare workers. 

It is an important study to help the decision makers to monitor the efficiency of control programs in the elimination of the KA. It would be as well important for other regions and countries to implement such a control program.

It demonstrates the use of the effort made to improve the deficiencies and to harmonize the level of qualifications at the different sentinel sites. 

It is however not clear which of these elements are prior in the reduction seen of the KA between 2011 to 2014.

Reviewer #2: This study addressed an important neglected disease which is Kala-azar. India representing 50% of the global case of the disease. The effort of the paper to explore possible interventions in terms of improving surveillance to identify Kal-azar cases as early as possible and to treat them make the paper of reasonable importance.

However, I have general and specific comments to the authors as followings:

(The line number referred to here is the line number of the word file of the manuscript)

- There have been a lot of abbreviations used through the manuscript, many of them used for the first time without full name. This would interrupt the focus of the reader. I would suggest you revise your abbreviations all over the paper and to write a additional table for your abbreviations so that the reader can check whenever needed.

- Abstract- you did not state clearly the objectives of the study.

- Abstract line 23, HMIS abbreviation appears for the first time, and has to be written in full name.

- Abstract, line 26, KA, this abbreviation appears for the first time, and has be written in full name.

- Abstract-methodology part- line 34, you need to define what do you mean by symptomatic cases ?

- Abstract- methodology part Line 46: SAG abbreviation appeared for the first time within the text and has to be written in full name.

- Abstract-principal findings part- you mentioned private health providers has low number of cases notification, how could we improve the private provider surveillance?- I suggest you to discuss this in some relevant part of the text.

Introduction- Line 96, is it manifest or manifests? grammar issue

Introduction- line 107- is it depend or depends? grammar issue.

Introduction, line 111, I think you mean counties and not countries, please check.

Material and methods- Line 147 to 151, reference is needed for this information.

Material and methods- Line 179- d is a typo.

Material and methods-line 243-246, you need a reference for the case definition of Kala-azar that you used.

Material and methods- line 248-250, you need a reference for the case definition of Post Kala-azar Dermal Leishmaniasis (PKDL) that you used.

Material and methods- Diagnosis of KA and PKDL, you need a reference to show that the tests suggested to use to identify patients with KA and PKDL are approved by national or international authorities such as WHO.

Material and method- Line 277- is there any reference for the standard protocol of Kala-azar and PKDL that you used in your study?

-Result-Line 518-Efficacy of anti kala-azar drugs administered at sentinel sites. Efficacy of drugs usually studied through Randomized Control Trails (RCT), which is not the case of your description here. Therefore, I would rather suggest you to rename this section with other title such as Response to anti kala-azar drugs administered at sentinel sites.

If you need to talk about the efficacy of your different protocols to treat Kala-azar then you have to carry out RCT.

Result- Line 544- ADR abbreviation used for the first time and has to be written in full name.

Discussion- Line 593, you need to put coma after India.

Discussion- Line 604, it should be have and not has, grammar issue.

Discussion- general comment that this part failed to explain possible explanations of the majority of the results stated by the authors as followings:

Discussion lines 612-618, you have only present other results compared to your results but you did not discuss the possible explanations for the seasonal variation of Kala-azar observed in your study.

Discussion part- Line 619- 630, you only mentioned the similarity and discrepancy of other studies compared to your result with regard to number of PKDL cases as double in Muzaffarpur and Samastipur district in the year 2013 when compared to 2011. However, you did not try to discuss the possible explanation for the high number of these two districts between 2011-2014.

Discussion, line 636-640, this is the first time that you give possible explanation for one of your results in the study namely gender differences of Kala-azar proportion.

- Discussion part-line 695, If the HMIS data was almost similar to the health facility sentinel surveillance data under the study of this manuscript, what is the need or advantage for this sentinel surveillance unless there are strong justifications need to be addressed by authors of this paper.

Conclusion- line 731, you advocated for the Health Facility-Based Sentinel Sites Surveillance System, my question how would such system will be sustained in the context of India, would you please elaborate on that at the text where you think is relevant.

Reviewer #3: The comments given in the preceding should be attended to before the manuscript is accepted for publication.

PLOS authors have the option to publish the peer review history of their article (what does this mean?). If published, this will include your full peer review and any attached files.

Reviewer #1: No

Reviewer #2: Yes: Osama Ahmed Hassan

Reviewer #3: Yes: Kalpana Pai
---

## [Decision Letter · Decision Letter 1]

2 May 2021

Dear Dr Rabidas,

Thank you very much for submitting your manuscript "Improved Kala-azar Case Management through Implementation of Health Facility-Based Sentinel Sites Surveillance System in Bihar, India" for consideration at PLOS Neglected Tropical Diseases. As with all papers reviewed by the journal, your manuscript was reviewed by members of the editorial board and by several independent reviewers. The reviewers appreciated the attention to an important topic. Based on the reviews, we are likely to accept this manuscript for publication, providing that you modify the manuscript according to the review recommendations. 

Please kindly recheck the manuscript for any grammatical errors and native editing. We need to sure your manuscript has an acceptable standard in terms of English language.

Sincerely,

Vahid Yazdi-Feyzabadi, PhD

Deputy Editor

Vahid Yazdi-Feyzabadi

Deputy Editor

Please kindly recheck the manuscript for any grammatical errors and native editing. We need to sure your manuscript has an acceptable standard in terms of English language.

Reviewer's Responses to Questions

**Key Review Criteria Required for Acceptance?**

**Methods**

-Are the objectives of the study clearly articulated with a clear testable hypothesis stated?

-Is the study design appropriate to address the stated objectives?

-Is the population clearly described and appropriate for the hypothesis being tested?

-Is the sample size sufficient to ensure adequate power to address the hypothesis being tested?

-Were correct statistical analysis used to support conclusions?

-Are there concerns about ethical or regulatory requirements being met?

Reviewer #2: (No Response)

Reviewer #3: The authors have taken the suggestions and improved the manuscript. 

The manuscript can now be accepted for publication.

**Results**

-Does the analysis presented match the analysis plan?

-Are the results clearly and completely presented?

-Are the figures (Tables, Images) of sufficient quality for clarity?

Reviewer #2: (No Response)

Reviewer #3: Yes the presentation of results comply with above points.

**Conclusions**

-Are the conclusions supported by the data presented?

-Are the limitations of analysis clearly described?

-Do the authors discuss how these data can be helpful to advance our understanding of the topic under study?

-Is public health relevance addressed?

Reviewer #2: (No Response)

Reviewer #3: Yes the conclusions are self explanatory.

**Editorial and Data Presentation Modifications?**

Reviewer #2: (No Response)

Reviewer #3: Accept

**Summary and General Comments**

Reviewer #2: The authors have responded appropriately to the comments.

Reviewer #3: The revised manuscript has improved.

PLOS authors have the option to publish the peer review history of their article (what does this mean?). If published, this will include your full peer review and any attached files.

Reviewer #2: Yes: Osama Ahmed Hassan

Reviewer #3: Yes: Kalpana Pai

Figure Files:

Data Requirements:

Reproducibility:

References

---

## [Editor Report · Decision Letter 2]

26 Jun 2021

Dear Dr Rabidas,

We are pleased to inform you that your manuscript 'Improved Kala-azar Case Management through Implementation of Health Facility-Based Sentinel Sites Surveillance System in Bihar, India' has been provisionally accepted for publication in PLOS Neglected Tropical Diseases.

Best regards,

Vahid Yazdi-Feyzabadi, PhD

Deputy Editor

Vahid Yazdi-Feyzabadi

Deputy Editor

---

## [Editor Report · Acceptance letter]

30 Jul 2021

Dear Dr Rabidas,

We are delighted to inform you that your manuscript, "Improved Kala-azar Case Management through Implementation of Health Facility-Based Sentinel Sites Surveillance System in Bihar, India," has been formally accepted for publication in PLOS Neglected Tropical Diseases.

Best regards,

Shaden Kamhawi

co-Editor-in-Chief

Paul Brindley

co-Editor-in-Chief
